# Structural Characterization and Biological Activity of Polysaccharides from Stems of *Houttuynia cordata*

**DOI:** 10.3390/foods11223622

**Published:** 2022-11-13

**Authors:** Xiaocui Liu, Jin Tian, Yinzhen Pan, Zhongqiao Li, Zhiran Zhou, Zihao Pan, Huazhang Tai, Yage Xing

**Affiliations:** 1Key Laboratory of Grain and Oil Processing and Food Safety of Sichuan Province, College of Food and Bioengineering, Xihua University, Chengdu 610039, China; 2Key Laboratory of Food Non Thermal Processing, Engineering Technology Research Center of Food Non Thermal Processing, Yibin Xihua University Research Institute, Yibin 644004, China

**Keywords:** *Houttuynia cordata* polysaccharide, extraction, structural characterization, biological activity

## Abstract

In this study, water-soluble natural polysaccharides were extracted from the stems of *Houttuynia cordata* Thunb (HCPS). The optimization of the hot water extraction process using response surface methodology (RSM), and the extraction factors, were analyzed by multiple stepwise regression analysis and Pearson analysis. Then, the structural characterization and biological activity of the HCPS were investigated. The results indicated that the maximum extraction yield (2.43%) of the HCPS was obtained at the optimal condition (extraction temperature for 90 °C, extraction time for 5 h, solid-liquid ratio for 1:30 g/mL). The extraction temperature was determined to be the primary factor influencing the extraction yield. The HCPS molecules had an average molecular weight of 8.854 × 103 kDa and were primarily of mannose (Man), rhamnose (Rha), glucuronic acid (GlcA), galacturonic acid (GalA), glucose (Glc), and xylose (Xyl). In addition, the backbone of the HCPS might consist of →6)-α-d-Glc*p*-(1→ and →6)-β-d-Gal*p*A-(1→. The HCPS had no triple-helix structure. The scanning electron microscopy (SEM) results showed that the HCPS presented a smooth and uniform appearance, and some sheet and chain structures existed. Moreover, the HCPS exhibited significant anti-oxidant activity and inhibited the activity of α-amylase and α-glucosidase. These findings showed that HCPS might be developed into a potential material for hypoglycemia, and provides a reference for the development of *Houttuynia cordata* polysaccharide applications in food.

## 1. Introduction

Diabetes mellitus (DM) is a metabolic disorder characterized by hyperglycemia, abnormalities in carbohydrate, fat, and protein metabolism, and chronic complications [1,2]. Type 2 diabetes mellitus (T2DM) had the highest incidence, followed by type 1 diabetes mellitus (T1DM) and gestational diabetes mellites [3]. The inhibition of α-glucosidase activity is an effective strategy for preventing the hydrolysis of carbohydrates into monosaccharides in the intestine, thereby mitigating T2DM and its complications [4,5]. Presently, synthetic α-glucosidase inhibitors are widely used in clinical settings to treat T2DM [6,7]. However, due to the adverse effects of certain synthetic drugs [8], there is a growing interest in seeking non-toxic, safe, and effective anti-diabetic candidates derived from natural sources. Due to their anti-diabetic activities, many natural polysaccharides have attracted considerable interest. Konjac glucomannan (KGM) is a polysaccharide isolated from the tubers of Amorphophallus konjac that can reduce the level of glucose in the blood [9]. Acanthopanax senticosus polysaccharide ASP was purified and found to significantly increase serum insulin levels while decreasing fasting blood glucose (FBG) levels [10]. Gracilaria lemaneiformis polysaccharide has the potential to reduce blood glucose disorder, MDA level, and pancreatic pathological damage [11]. Thus, polysaccharides extracted from natural resources have a great potential application in the development of novel pharmaceuticals and functional food ingredients for T2DM treatment and prevention.

*Houttuynia cordata* Thunb. (*H. cordata*), as traditional herbal medicine, is a single species of its genus and is widely distributed in Southeastern and Northeastern Asian countries [12]. The Chinese discovered the medicinal properties of *H. cordata* and have used it for thousands of years in traditional Chinese medicine. The green leaves and young roots of *H. cordata* are also popular vegetable and agricultural products in Southern China today. The pharmacological study demonstrated that *H. cordata* extracts possess multiple bioactivities, such as anti-cancer [13,14], anti-obesity [15,16], anti-inflammatory [17], anti-allergic [18], anti-oxidative [19], anti-viral [20], and immunomodulatory effects [21]. Polysaccharides extracted from the natural resources of *H. cordata* have increased in popularity due to their extensive use in disease treatment. HCP-2 from *H. cordata* had immunomodulatory activity, was primarily composed of galacturonic acid, and was elucidated as linear poly -(1→4)-α-d-galactopyranosyl uronic acid by Cheng et al. [22]. Cheng et al. [23] reported that *H. cordata* (HP) had an anti-viral effect. It was composed of galacturonic acid, galactose, glucose, and xylose, with the glycosidic bond types consisting primarily of α-1,4-linked GalpA, β-1,4-linked Galp, β-1,4-linked Glcp, and β-1,4-linked Xylp. Han et al. [24] found that the main components of the *H. cordata* polysaccharide (HCA4S1) are rhamnose, galacturonic acid, galactose, and arabinose, which are formed by 1,4-linked α-d-GalA and 1,2,4-linked α-l-Rha. HCA4S1 might inhibit lung cancer cell proliferation in A549 patients. In the North Eastern states of India, it is used as a garnish on ethnic side dishes and leaf salads to reduce blood sugar levels [25]. The activity of *H. cordata* polysaccharide has been extensively studied, as shown in the Table 1. Nevertheless, the effect of *H. cordata* polysaccharides on α-amylase and α-glucosidase activity is unknown. This restricts the medical and efficient applications of *H. cordata* agricultural resources.

In this work, the extraction method of a one novel heteropolysaccharide was optimized from *H. cordata* by response surface methodology and purified by Sephadex G-200. Its chemical structure was characterized. Three antioxidant assay methods evaluated its antioxidant activity. The inhibitory activity of *H. cordata* polysaccharides on α-amylase and α-glucosidase in vitro were explored to expand *H. cordata* polysaccharide utilization in the field of medicine.

## 2. Materials and Methods

### 2.1. Materials and Chemicals

*H. cordata* was purchased from Wuxi Xiaogu e-commerce Co., Ltd. (Chongqing, China). The materials were thoroughly washed with tap water, air-dried at room temperature, and then finely powdered with a pulverizer through 30 mesh sieves. d-glucose, l-rhamnose, d-xylose, d-mannose, d-glucuronic acid, d-galacturonic acid, and Sephadex G-200 were purchased from Sigma Chemical Co., Ltd. (St. Louis, MO, USA). Dextran (T-10, T-40, T-70, T-500, and T-2000) was obtained from Solarbio Company (Beijing, China). All other chemicals used in this study were of analytic grade.

### 2.2. Extraction, Isolation, and Purification of Polysaccharides

#### 2.2.1. Extraction of HCPS

The HCPS extraction referenced by Lu et al. [30] was modified. The following conditions were optimized using a single-factor and an RSM experiment. HCPS extraction rate optimization was investigated using a single-factor experiment with different solid-liquid ratio (1:20, 1:30, 1:40, 1:50 and 1:60 g/mL), extraction times (2, 3, 4, 5 and 6 h), and temperatures (60, 70, 80, 90, and 100 °C).

The RSM experiment optimized extraction time (X_1_), extraction temperature (X_2_), and solid-liquid ratio (X_3_). Table 2 presents the three-level variables for each variable.

#### 2.2.2. Multiple Stepwise Regression Analysis of Extraction Factors

The extraction yield of HCPS was affected differently by extraction time, temperature, and solid-liquid ratio, but the influence was different. To provide a theoretical foundation for HCPS extraction, we utilized multiple stepwise linear regression to analyze the mathematical relationship between influencing factors and to develop a regression equation model that accurately predicted the factors that had the greatest influence on the HCPS extraction rate.

#### 2.2.3. Pearson Correlation Analysis of Extraction Factors

Pearson correlation coefficient was used to measure the linear correlation between two variables in statistics, and its value ranges between −1 and 1. The two variables are positively correlated and can be expressed by a linear equation if the value is 1. Due to their negative correlation (as demonstrated by the coefficient value of −1), these two variables can be expressed by a linear equation. The closer the absolute value of the coefficient is to 0, the weaker the linear correlation between two variables. The higher the linear correlation between two variables, the closer the absolute value of the coefficient is to 1.

#### 2.2.4. Isolation and Purification of HCPS

In the extract, excess anhydrous ethanol precipitated polysaccharides. The crude HCPS was obtained by deproteinizing the retentate portion using the Sevage (1-butanol/chloroform 1:4, *v*/*v*) method and lyophilization it after dialysis (cut-off MW 10,000 Da).

The polysaccharide sample was scanned at all wavelengths to detect the presence of protein residues. Sugar content was determined by treating crude polysaccharides with phenol sulfuric acid. High-performance gel permeation chromatography (HPGPC) was used to detect the various crude polysaccharide distributions. The polysaccharide was gel filtered on a column (30 cm × 3 cm) of Sephadex G-200, eluted with deionized water at a flow rate of 0.3 mL/min, and monitored using the phenol-sulfuric acid method. Finally, HCPS was obtained.

### 2.3. Homogeneity and Molecular Weight Determination

The distribution of polysaccharide molecular weight (*M_w_*) was determined using an HPGPC (Agilent-1200, Santa Clara, CA, USA) equipped with a TSK gel G4000 PWxl column (7.8 mm × 300 mm, column temperature 30 °C) and Refractive Index Detector (RID, detecting temperature 35 °C). A sample solution (20 µL) was injected and run at 0.6 mL/min with deionized water as the mobile phase. The standard curve was created using T-series Dextran as the standard (T-10, T-40, T-70, T-500 and T-2000) [31]. 

### 2.4. Monosaccharide Identification

The polysaccharides were hydrolyzed in a sealed tube with 2 M TFA for 3 h at 110 °C. A part of the hydrolysate was subjected to TLC analysis following the removal of TFA. The other part was acetylated and analyzed by HPLC [32]. Derivatized standards included d-glucose, l-rhamnose, d-xylose, d-mannose, d-glucuronic acid, and d-galacturonic acid.

### 2.5. FT-IR and NMR Analyses

For the analysis, 1 mg of the sample was mixed with 150 mg of dry KBr and pressed into a 1 mm thick disk. On a Bruker Vector22 Fourier transformed IR spectrophotometer (Billerica, MA, USA), the IR spectrum was recorded in the absorbance mode in the range of 4000–400 cm^−1^ (VECTOR-22) [33]. At a probe temperature of 298 K, ^1^H NMR and ^13^C NMR spectra were recorded on a Bruker spectrometer (Zurich, Switzerland) (600 MHz). The freeze-dried samples were measured after being exchanged twice with D_2_O, respectively [34]. 

### 2.6. Periodate Oxidation 

The periodate oxidation analysis was conducted using methods from the literature [35]. Preparation of standard curve of sodium periodate consumption: prepared 30 mmol L^−1^ sodium periodate solution, drew the standard curve with the gradient concentration of sodium periodate as the abscissa and the OD value of absorbance as the ordinate. Sample determination: added sodium periodate solution to the polysaccharide sample, reacted at room temperature in the dark, and measured the absorbance value at 223 nm every 8 h. When the absorbance reached a stable value, added ethylene glycol to stop the reaction. From this absorbance value, the consumption of periodate could be calculated according to the standard curve. Periodate oxidation standard curve was as follows:Y = 0.0102X − 0.0008   R^2^ = 0.9998(1)

### 2.7. Methylation Analysis

In distilled water (10 mL), 20 mg of the sample was dissolved. 1-(3-dimethylaminopropyl)-3-ethylcarbodiimide hydrochloride was combined with the solution (0.5 g). First, the pH was adjusted to 4.75 with HCl (0.1 mol·L^−1^); then NaBH_4_ (2 mol·L^−1^) was added slowly, and the reduction continued for 2 h. The pH of the system was maintained at around 7.0 using HCl (4 mol·L^−1^). After the reaction, the solution was dialyzed for 48 h using a dialysis bag (3500 D) and then concentrated and lyophilized. The lyophilized sample was then dialyzed and lyophilized again.

The lyophilized sample was methylated according to the method described in the literature [36]. Briefly, the lyophilized sample was dissolved in dry DMSO (2 mL), and fresh NaOH powder was added (25 mg); then, the mixture was sonicated for 30 min. Gradually, methyl iodide was added to the solution, which was then sonicated in the dark for 2 h. Distilled water decomposed the remaining methyl iodide. Pre-methylated products were extracted with CH_2_Cl_2_. The extract was dried with N_2_, yielding a methylated sample. IR was used to detect the degree of methylation. Following successive methylation, the sample was dissolved in TFA (2 mL, 2 mol·L^−1^) and hydrolyzed at 110 °C for 3 h. To remove TFA the hydrolyzed sample was dried by Heidolph rotary vacuum evaporation (Schwabach, Germany) and co-distilled three times with methanol. The hydrolyzed sample was dissolved in deionized water (2 mL); then, NaBH_4_ (25 mg) was added to reduce the hemiacetal groups. The reduction was terminated with acetic acid. Three rounds of co-distillation with methanol and rotary vacuum evaporation were used to dry the sample. The sample was acetylated with acetic anhydride and analyzed for partially methylated alditol acetates by GC-MS. GC-MS chromatographic conditions: gas chromatography-mass spectrometer (GC-MS) equipped with a VF-5 ms column (30 m × 0.25 mm × 0.25 μm). Column flow (mol·L^−1^): 0.9. Column pressure (psi): 12.2. Column oven temp: 320 °C. Scanning time: 20 min.

### 2.8. Congo-Red Test

The Congo red interaction with polysaccharides was measured by its visible absorption maximum. The sample solution (0.5 mg/mL) was combined with 2 mL, 50 μmol/L Congo red, and 1 M NaOH to adjust the final concentration of NaOH to 0, 0.05, 0.10, 0.15, 0.20, 0.25, 0.30, 0.35, and 0.40 M. After holding the samples at room temperature for 10 min, the UV-Vis absorption spectrum was recorded on Shimadzu UV-1900 spectrophotometer (Kyoto, Japan) (400–600 nM) [37].

### 2.9. Scanning Electron Microscopy (SEM)

The samples were fixed onto a copper stub, respectively. After sputtering with a layer of gold, the samples were examined with a Hitachi SU1510 scanning electron microscope (Tokyo, Japan) [38].

### 2.10. In Vitro Anti-Oxidant Activity 

#### 2.10.1. The Scavenging Efficiency of DPPH •

The scavenging efficiency of complexes for DPPH • was determined using the methods of Cheng et al. (2019) [39] used vitamin C as control and mixed 2 mL of the complex with 0.1 mM DPPH • the same volume of solution, thoroughly mixed. The system reacted in a dark place for 30 min and then measured at 510 nm. The sample and DPPH • solutions were replaced with deionized water and absolute ethanol, respectively. The calculation formula was as follows:(2)$$\mathrm{Scavenging}\;\mathrm{Efficiency}\;(\%)=\frac{A_{0}{\;-\;(A}_{1}{\;-\;A}_{2})}{A_{0}}\;\times \;100$$
where A_0_ is deionized water and DPPH • solution, A_1_ is complexes and DPPH • solution, and A_2_ is complexes and absolute ethanol.

#### 2.10.2. The Scavenging Efficiency of •OH

Complexes were thoroughly mixed with FeSO_4_ solution (9 mmol/L) and hydrogen peroxide solution (6 mmol/L) according to Tian et al. [19]. The above mixture was added to a solution of salicylic acid (9 mmol/L) at 37 °C under a constant temperature condition for 10 min and then reacted under the condition of avoiding light for 30 min. Finally, the above mixture was measured at 600 nm. The calculation formula was as follows:(3)$$\mathrm{Scavenging}\;\mathrm{Efficiency}\;(\%)=\frac{A_{0\;}{-\;(A}_{1}{\;-\;A}_{2})}{A_{0}}\;\times \;100$$
where A_0_ is deionized water + FeSO_4_ + H_2_O_2_ + salicylic acid solution, A_1_ is complexes + FeSO_4_ + H_2_O_2_ + salicylic acid solution, A_2_ is complexes + FeSO_4_ + deionized water + salicylic acid solution.

#### 2.10.3. The Scavenging Efficiency of ABTS^+^

The ABTS^+^ scavenging ability was determined according to the method of de Falco et al. [40]. ABTS and potassium persulfate were mixed and stored at 4 °C. Before use, the stock solution of ABTS^+^ should be diluted to an OD of 0.7–0.8. Mixed the sample with ABTS^+^ solution, reacted for 3 min, and measured at 734 nm. Deionized water served as the blank control sample. The calculation formula was as follows:
(4)$$\mathrm{Scavenging}\;\mathrm{Efficiency}\;(\%)=\;\frac{{\mathrm{OD}}_{\mathrm{control}}{\;-\;\mathrm{OD}}_{\mathrm{sample}}}{{\mathrm{OD}}_{\mathrm{control}}}\;\times \;100$$
where OD_control_ is blank control, OD_sample_ is sample.

### 2.11. In Vitro α-Amylase Inhibitory Activity Assay

The polysaccharide concentrations were 2.0, 4.0, 6.0, 8.0, 10.0 mg/mL. PBS (pH = 6.8) and α-amylase was added to the sample in the same volume and incubated for 10 min at 37 °C. Then, a water bath was maintained at 37 °C with the starch solution for 20 min. Finally, the sample was measured at 540 nm after adding DNS to terminate the catalytic reaction [41]. The inhibitory rate was calculated as follows:Inhibition rate (%) = [1 − (A_1_ − A_2_)/(A_3_ − A_0_)] × 100
(5)
where A_0_ is deionized water + PBS + starch solution, A_1_ is sample + PBS + α-amylase + starch solution, A_2_ is sample + PBS + deionized water + starch solution, A_3_ is deionized water + PBS + α-amylase + starch solution.

### 2.12. In Vitro α-Glucosidase Inhibitory Activity Assay

The ELISA plate served as the reaction carrier for the assay, which was performed according to the method of Ren Yuanyuan [42]. The concentration of polysaccharide was 2.0, 4.0, 6.0, 8.0, 10.0 mg/mL, respectively. The substrate solution was prepared with PBS (pH = 6.8). In the sample group, 40 μL of 0.2 U/mL α-glucosidase and sample (40 μL) solution were added to each hole, whereas the background group received the same amount of PBS instead of the sample. The blank group received only an equivalent amount of α-glucosidase and PBS solution. After a constant water bath at 37 °C for 10 min, 40 μL of 2.5 mmol/L PNPG was added. After 30 min of reaction time at 37 °C, 0.2 mol/L Na_2_CO_3_ (120 μL) was added to terminate the catalytic reaction. Optical density (OD) at 405 nm was used to quantify the enzyme activity. All assays were conducted in triplicates. The inhibitory rate of sample on α-glucosidase was calculated by the following formula.
Inhibition rate (%) = (OD_control_ − OD_sample_)/(OD_control_ − OD_blank_) × 100%(6)

### 2.13. Statistical Analysis

All results are expressed in the tables as mean ± standard error (SE). The differences between groups were determined using the Tukey test and SPSS 17.0 (Amunk, New York, USA). A probability value of *p* less than 0.05 was considered significant.

## 3. Results

### 3.1. Polysaccharide Extraction

#### 3.1.1. Single-Factor Experiment

As shown in Figure 1A, the extraction yield of HCPS was highest at 1:30. After exceeding 1:30, the extraction yield of the HCPS decreased with the increase of the solid-liquid ratio. As a result of increasing the amount of solvent, the contact area between the HCPS and the solvent increased, thereby enhancing the mass transfer efficiency [43]. However, when there were too many solvents, most of the HCPS dissolved into the solvent to form a saturated solution, and the HCPS was difficult to dissolve. The extraction yield of the HCPS increased with the increasing temperature but gradually decreased when the temperature exceeded 90 °C, indicating that the HCPS was sensitive to temperature. Due to the increase in temperature, the HCPS accelerated the rate of migration to the solvent. When the temperature was too high, the browning and degradation of the HCPS hampered its extraction [44]. The extraction yield of the HCPS increased from 2 h to 5 h as the extraction time increased. This may be due to the low initial concentration of crude polysaccharides in the solvent and the diffusion of polysaccharide components from the raw materials into the solvent. Due to the degradation or inactivation of polysaccharide components after 5 h of extraction, the extraction yield of the HCPS gradually decreased as time progressed. The optimal extraction conditions were investigated at 1:30 g/mL at 90 °C for 5 h.

#### 3.1.2. RSM Experiment

The RSM test was used to optimize the optimal conditions for the HCPS extraction as determined by the single factor test, and the results were analyzed by regression. The results are shown in Table 3 and Table 4. According to the regression analysis, the regression equation of the extraction yield of HCPS was as follows:Y(%) = 2.35 + 0.13X_1_ + 0.16X_2_ − 0.0088X_3_ + 0.087X_1_X_2_ − 0.0025X_1_X_3_ + 0.055X_2_X_3_ − 0.27X_1_^2^ − 0.28X_2_^2^ − 0.20X_3_^2^
(7)
where Y is extraction yield; X_1_ is extraction time; X_2_ is extraction temperature; X_3_ is solid-liquid ratio.

The *p* value of the model was 0.0115 (*p* > 0.05) according to the analysis of variance, indicating that the model is significant. The lack of fit was not statistically significant (*p* > 0.05), so the model selection was appropriate. X_1_, X_2_ and X_1_^2^, X_2_^2^, and X_3_^2^ were significant, and X_1_^2^ and X_2_^2^ were highly significant to the results. According to the F-test, these three variables influenced the extraction yield in the following order: X_2_ (extraction temperature) > X_1_ (extraction time) > X_3_ (solid-liquid ratio). The R-Squared is 89.16% > 80%, indicating that the measured and predicted values were well fitted, and the model could be used for prediction. This model could truly reflect the test results due to the adeq precision = 6.58 > 4.

The 3D response surface plots (Figure 1B) intuitively reflected the influence of variables on the extraction yield and the interaction of variables, which is identical to the outcome of variance analysis. The optimal extraction conditions could be obtained as follows, based on the above analysis: extraction temperature, 93.44 °C; extraction time, 5.29 h; solid-liquid ratio, 1:30.25 g/mL; extraction yield, 2.40%. For the feasibility and simplicity of practical operation, the extraction conditions were revised: extraction temperature of 90 °C, extraction time of 5 h, and the solid-liquid ratio of 1:30 g/mL. The extraction yield was 2.43% (n = 3), which is consistent with the predicted value.

#### 3.1.3. Multiple Stepwise Regression Analysis

The purpose of multiple regression analysis is to examine the impact of multiple factors on a single object. It is usually expressed by y, and the influencing factors are named by independent variables. It is usually expressed by a vector (x_1_, x_2_, x_3_,--- x_p_). If there is a linear relationship between these vectors and the object, these relationships can be described by the following expression for multiple linear regression [45].
y = a_0_ + b_1_ x_1_ + b_2_ x_2_ + … +b_p_ x_p_(8)

Multiple linear stepwise regression was utilized to study the HCPS extraction process. The residual statistics showed that the maximum cook = 0.94 < 1, and the data met the requirements for multiple stepwise regression analysis. According to Table 5, the extraction temperature first entered the model creating model 1. Extraction time was then entered into the model and model 2 was established. The solid-liquid ratio was excluded from the regression model (*p* > 0.05). Because model 1 R^2^ < model 2 R^2^, the prediction results of model 2 are more consistent with the actual situation. Both models of regression are statistically significant (Table 6) (*p* < 0.05).

The constant terms of the two models were not significant, as evidenced by Table 7. As such, the regression equation of model 1 was Y = 0.794X1, and the regression equation of model 2 (extraction time = X_1_, extraction temperature = X_2_) was Y = 0.683X_1_ + 0.487X_2_. In model 2, the temperature standardization coefficient was greater than the time standard coefficient, indicating that temperature plays a significant role in HCPS extraction. Model 1 and 2 tolerance and VIF are both 1, showing that the data are consistent with multiple stepwise regression analysis. There were no multicollinearity issues with extraction temperature and time. In conclusion, the extraction temperature had the greatest impact on the extraction of the HCPS, followed by the extraction time, and the solid-liquid ratio had little impact on the extraction of the HCPS.

#### 3.1.4. Pearson Correlation Analysis

There was a significant linear relationship between the extraction temperature and extraction yield. The medium linear relationship between the extraction time and extraction yield was insignificant. Their correlation between the solid-liquid ratio and the extraction yield was weak and not statistically significant (Table 8). Therefore, the extraction temperature was the primary factor affecting the extraction yield, similar to the multiple stepwise regression analysis.

### 3.2. Isolation and Purification of Polysaccharides

The *H. cordata* stem was degreased, extracted with hot water, precipitated with ethanol, dialyzed, and deproteinized to isolate the HCPS. Then, the HCPS was purified using the molecular sieve effect of Sephadex G-100, yielding 4.0% (*w*/*w*), as shown in Figure 2A.

### 3.3. Homogeneity and Molecular Weight Determination of HCPS

In HPGPC, the HCPS exhibited a single and symmetric peak, declaring that the HCPS was a homogeneous polysaccharide (Figure 2B). The molecular weight determination of the HCPS was 8.854 × 10^3^ kDa. The UV absorption spectrum of the HCPS revealed no absorption peak at 260–280 nm, indicating the absence of protein (Figure 2C).

### 3.4. Structure Characterization

#### 3.4.1. Monosaccharide Composition of HCPS

The presence of Man (4.22%), Rha (24.75%), GlcA (3.67%), GalA (10.42%), Glc (30.42%), and Xyl (20.55%) in the HCPS indicated that it was an acidic polysaccharide. Xu et al. [46]. isolated the polysaccharide (HCP) from *H. cordata*, which contained the monosaccharides Glc, Gal, Ara, and Rha as their primary constituents. HCP differed from the monosaccharide composition of this study, which may be due to differences in the source of the raw materials.

#### 3.4.2. FT-IR Spectrometric Analysis

Figure 3A shows the typical characteristic absorption peaks of polysaccharides. The typical polysaccharide absorption peak at 3422 cm^−1^ corresponds to the stretching vibration of the O-H bond in the polysaccharide functional group [47]. The stretching vibration of C-H in polysaccharide functional groups corresponds to 2928 cm^−1^ [48]. The absorption peaks of the HCPS at 1621 cm^−1^ and 1424 cm^−1^ indicate that it contains uronic acid [49]. The absorbance peak band at 1100 cm^−1^ and 1028 cm^−1^ confirmed the presence of Pyranose (C–O–C and C–O–H) [50]. The absorption peak of 892 cm^−1^ and 832 cm^−1^ indicated that the HCPS possessed β-glycosidic and α-glycosidic bonds [51,52]. The HCPS was identified as an acidic polysaccharide by the absorption peak at 1243 cm^−1^ (C–O and/or C–N) [53].

#### 3.4.3. NMR Analysis

NMR was used to analyse the HCPS structure further. The anomer hydrogen of polysaccharide molecule is α-configuration in the range of 5.0–6.0 ppm, while β-anomeric protons occur at 5.0–4.0 ppm [54]. Eight chemical shift signals appeared on the ^1^H-NMR (Figure 4A) at δ 5.08 ppm, δ 4.42 ppm, δ 4.10 ppm, δ 3.98 ppm, δ 3,78 ppm, δ 3.32 ppm, δ 1.88 ppm, and δ 1.22 ppm. It indicated that the glycosidic bond type of the HCPS contained both α- and β-configurations, simultaneously. As seen in Figure 4B, there were six signal peaks in the ^13^C-NMR spectrum of HCPS (23.29 ppm, 48.90 ppm, 67.81 ppm, 70.59 ppm, 99.69 ppm and 170.78 ppm). The signal peak intensity of 170.78 ppm was related to C-6 of 1,4-α-d-GalAMe*p*. The signals at 1.22 ppm and 4.10 ppm were assigned to H-2 and h-6 of 1,2-α-l-Rha [55]. The signal at 48.90 ppm illustrated the presence of methoxyl groups, whereas the H/C signals of the acetyl groups appeared at 1.88/22.29 ppm. The peaks at 5.08/99.69 ppm were derived from H-1/C-1 of →6)-α-d-Glc*p*-(1→ [56]. The signals at 3.78/67.81 ppm and 3.98/70.59 ppm were clearly assigned to H-2/C-2 and H-3/C-3 of →6)-β-d-Gla*p*-(1→ [22,23,24]. The resonance signals of the HCPS in ^1^H-NMR and ^13^C-NMR spectrum were crowded and difficult to identify, due to the large molecular weight of the HCPS. Additionally, the content of some monosaccharide residues in the HCPS was low, so it was not possible to show all heterocephalic carbon signals. Han et al. [24] extracted polysaccharide (HCA4S1) from *Houttuynia cordata* Thunb. The glycosidic bond was mainly 1,4-linked-α-d-GalA; 1,2,4-linked-α-l-Rha. The glycosidic bond of *Houttuynia cordata* polysaccharide (HCP-2) extracted by Cheng et al. [22] was mainly 1,4-linked-α-d-GalA. The main glycosidic bond of *Houttuynia cordata* polysaccharide (HBHP-3) extracted and purified by Zou et al. [28] was → 2)-α-l-Rha*p*-(1→, →4)-α-d-Gal*p*A—(1→ and →4)-β-d-Gal*p*-(1→. These are not similar to HCPS, so *Houttuynia cordata* polysaccharide had different structures due to different place of origin and extraction of different parts.

#### 3.4.4. Periodate Oxidation

Periodate can selectively break the adjacent dihydroxy or Ortho trihydroxy junction in the polysaccharide structure, resulting in the formation of polysaccharide aldehyde, formaldehyde, or formic acid. A molecule of periodate must be consumed after an A-C-C bond is oxidized. After periodic acid oxidation, different connection modes between monosaccharides can produce different oxidation products. Therefore, the position and proportion of glycosidic bonds in polysaccharides can be calculated based on the consumption of periodic acid, the production of formic acid, and the amount of residual sugar residues [57]. The consumption of sodium periodate in the HCPS was 0.75 mmol, and the production of formic acid in the HCPS was 0.011 mmol. This indicated that the HCPS had 1→or 1→6 bonded glycosyl. Meanwhile, the HCPS also had 1→2or 1→4 bonded glycosyl. In addition, the amount of periodate consumed per mole of glucose residue was less than 1 mol, indicating that the HCPS could contain 1→3 bonded glycosyl because the 1→3 bond did not consume periodate in the oxidation process [58].

#### 3.4.5. Methylation Analysis

Figure 3B shows the FTIR spectroscopy of the HCPS after methylation treatment. Significant enhancement of the methyl characteristic absorption peak around 2900 cm^−1^ indicates complete methylation. PMAAs were determined using GS-MS after completely methylated the HCPS was hydrolyzed, reduced, and derivatized. Methylation analysis revealed that the HCPS contained three glycosidic linkages, including →6)-Glc*p*-(1→, →4)-Man-(1→and →6)-Gal*p*A-(1→. In the study of Zhou et al. [28], HBHP-3 was isolated and purified from *Houttuynia cordata* Thunb. HBHP-3 was composed of eight glycosidic bonds, including 1-Ara*f*, 1,5-Ara*f*, 1,2-Rha*p*, 1,2,4-Rha*p*, and 1,4-Glc*p*, et al. The methylation analysis result of HCPS was different from that of HBHP-3, which might be caused by the different monosaccharide composition. The low content of the HCPS may have resulted in a low number of peaks in GS-MS, explaining the low number of glycosidic bonds detected.

#### 3.4.6. Congo-Red Test

Congo-red can bind to polysaccharides with three-strand helical structure, within a specific NaOH concentration, the maximum absorption wavelength of the complex solution is significantly red-shifted compared to pure Congo-red [59]. As shown in Figure 5A, in 0.05~0.4 mol/L, Congo-red—HCPS complex solution maximum absorption wavelength increased then stabilized. Although the maximum absorption wavelength of Congo-red-HCPS complex solution was significantly higher than that of the pure Congo-red group in the concentration range of 0.05~0.4 mol/L NaOH, there was no significant red shift in the maximum absorption wavelength of Congo-red—HCPS complex solution when compared to the pure Congo-red group. The HCPS lacked a three-strand helical chain conformation, while the maximum absorption wavelength of Congo-red-HCPS complex solution remained unchanged in high concentration NaOH solution. Therefore, this may be due to the higher molecular weight and hyperbranched structure of the HCPS. 

#### 3.4.7. Scanning Electron Microscopy (SEM) Analysis

The SEM images of the HCPS are shown in Figure 5B,C. The HCPS displayed complex entanglement structures, primarily sheet and chain structures. The surface of the HCPS was smooth and uniform. It could be seen from the microstructure that there were a lot of voids on the surface of the HCPS and obvious spaces inside. The reticular structure exposed more active regions of the HCPS, which was conducive to improving biological activity [60]. *Houttuynia cordata* polysaccharide was composed of different thin slices and it was similar to the HCPS structure, according to Liu et al. [26].

### 3.5. Anti-Oxidant Activity In Vitro

As shown in Figure 6A, the scavenging activities of the HCPS on DPPH•, •OH, and ABTS^+^• were correlated positively with its concentration. At the concentration of 1.6 mg/mL, the clearance rates of DPPH•, •OH, and ABTS^+^• were 43.26%, 26.68%, and 16.89%, respectively, but the clearance rates of these were lower than those of the positive control. Previous studies have shown that the anti-oxidant activity of polysaccharides was related to their molecular weight, monosaccharide composition, and a viscosity [61,62]. *Houttuynia cordata* polysaccharide (HCP) was composed of mannose, rhamnose, glucuronic acid, galacturonic acid, glucose, xylose, galactose, and arabinose, according to the study of Tian et al. [19]. The clearance rates of DPPH• and •OH of HCP were 70.2% and 57.7%, respectively, which are higher than those of the HCPS studied in this paper. Different monosaccharide compositions and different raw material origins may cause this difference. The low anti-oxidant activity of the HCPS was related to the high molecular weight, as low molecular weight polysaccharides had more hydroxyl, a larger surface area, and simple contact with free radicals [63,64].

### 3.6. α-Amylase and α-Glucosidase Inhibitory Activity

α-amylase can convert carbohydrates (such as starch) into oligosaccharides, whereas α-glucosidase increases blood glucose levels by preventing the hydrolysis of oligosaccharides into glucose [65,66]. Therefore, it can effectively delay the release of glucose by lowering postprandial blood glucose levels and inhibiting α-amylase and α-glucosidase, which play a hypoglycemic role [67]. Figure 6B shows the inhibitory effect of the HCPS on α-amylase and α-glucosidase activities. The α-amylase and α-glucosidase activities were inhibited by the HCPS in a dose-dependent manner. The inhibitory activities of α-amylase and α-glucosidase increased with the increasing concentration; at a concentration of 10 mg/mL, the maximum inhibitory activities of α-amylase and α-glucosidase were respectively 34.31% and 18.43%. Previous studies have presented that the inhibitory activities of polysaccharides on α-amylase and α-glucosidase were closely related to their structural characteristics [68]. Chen et al. [69] indicated that the corn silk native polysaccharide (N-CSPS) inhibited α-amylase activity. N-CSPS primarily consisted of Glc, Gal, Man, and Ara. Wang et al. [70] enunciated that the inhibitory activity of polysaccharide WAFP from wax apple on α-glucosidase, Rha, Ara, Xyl, Man, Glc, Gal, and GalpA made up WAFP and possessed→4)-α-d-Glcp-(1→, →3,4)-β-d-Xylp-(1→ and →3)-β-d-Galp-(1→. In this study, the HCPS was composed of Rha, GlcA, Xyl, Man, Glc, and possessed GalA→6)-α-d-Glc*p*-(1→ and →6)-β-d-Gal*p*-(1→. Nevertheless, for α-amylase and α-glucosidase, the inhibitory activity of the HPCS was not particularly strong; it needed to be improved. Dou et al. [71] explained that the degradation of blackberry fruit polysaccharide could effectively improve its inhibitory activity on α-glucosidase. In general, the HCPS can inhibit α-amylase and α-glucosidase. Other methods to improve the HCPS inhibitory activity on α-amylase and α-glucosidase must be studied, and the inhibitory effect of the HCPS on α-amylase and α-glucosidase can be explored further, such as through in vivo experiments.

## 4. Conclusions

This study determined the optimal conditions for extracting polysaccharides from *Houttuynia cordata* stems (HCPS) using response surface methodology, the optimal extraction process of the HCPS was as follows: the extraction time of 5 h, the extraction temperature of 90 °C, the solid-liquid ratio of 1:30, and had the extraction rate of 2.43 ± 0.12%. The HCPS was an acidic polysaccharide with Man, Rha, GlcA, GalA, Glc, and Xyl, and consisted of →6)-α-d-Glc*p*-(1→ and →6)-β-d-Gal*p*-(1→, and the molecular weights was determined to be 8.854 × 10^3^ kDa. Congo red test showed that the HCPS did not have triple helix configuration. The results of the biological activity test suggested that the HCPS was capable of scavenging DPPH•, •OH, and ABTS^+^•, the HCPS also might inhibit α-amylase and α-glucosidase. Therefore, this polysaccharide may be developed as a potential material for hypoglycemia. Further studies are indispensable to discover the relationship between the structure and bioactivity of the HCPS.

## Figures and Tables

**Figure 1 foods-11-03622-f001:**
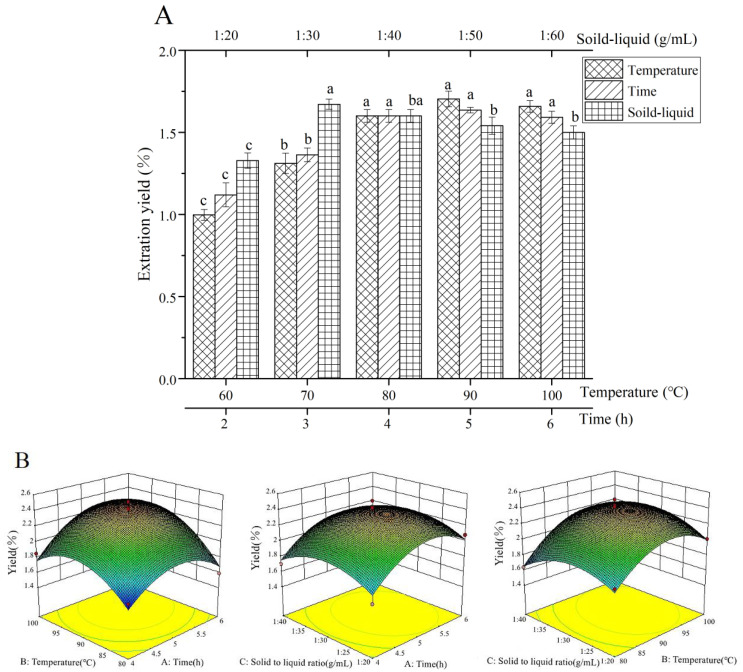
Results of single factor and RSM, (**A**): single factor, (**B**): RSM. Different lower-case letters were different significantly (*p* < 0.05).

**Figure 2 foods-11-03622-f002:**
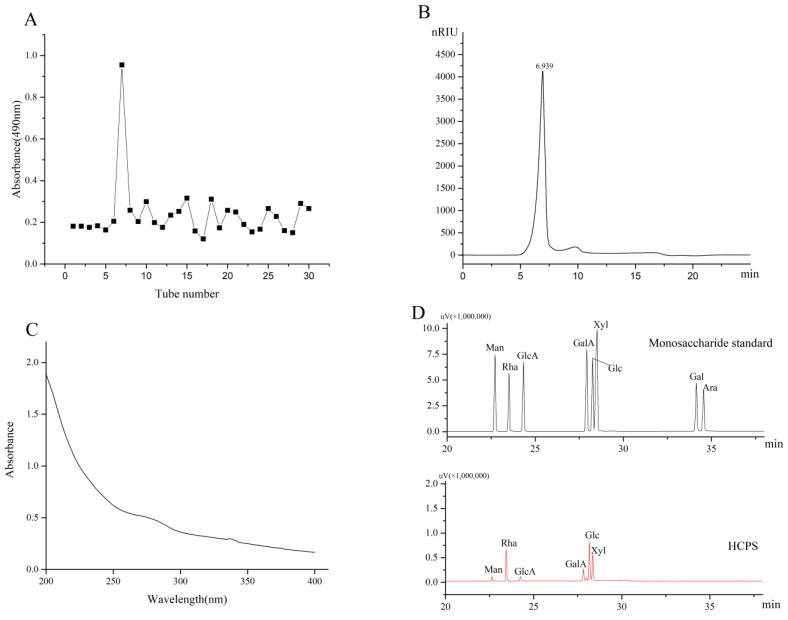
(**A**) Chromatograms of HCPS, (**B**) HPGPC of HCPS, (**C**) UV of HCPS, (**D**) HPLC profiles of acid degraded HCPS and monosaccharide standards.

**Figure 3 foods-11-03622-f003:**
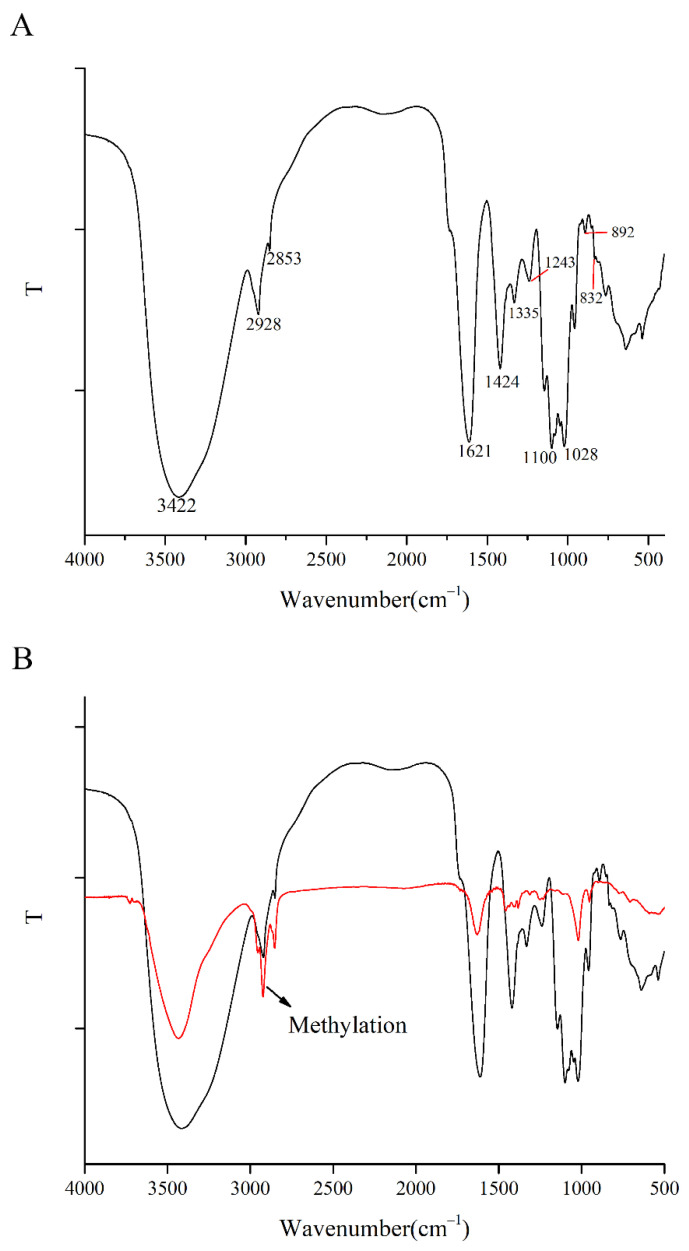
(**A**) IR spectrum of HCPS, (**B**) IR spectrum of methylated HCPS.

**Figure 4 foods-11-03622-f004:**
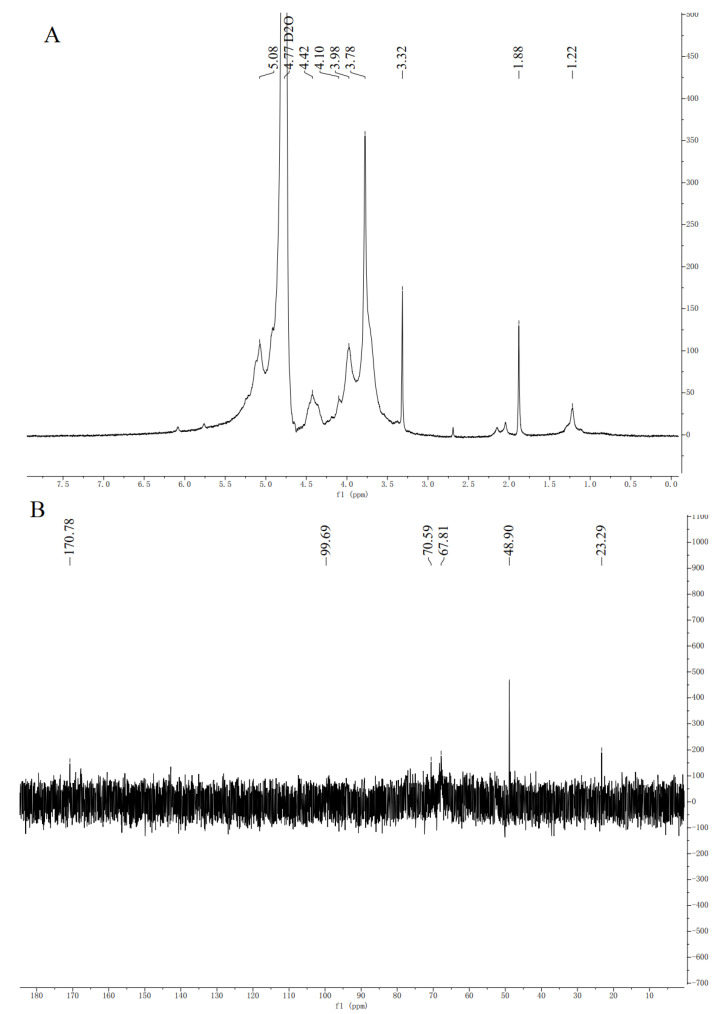
(**A**) The ^1^H NMR spectra of HCPS, (**B**) ^13^C NMR spectra of HCPS.

**Figure 5 foods-11-03622-f005:**
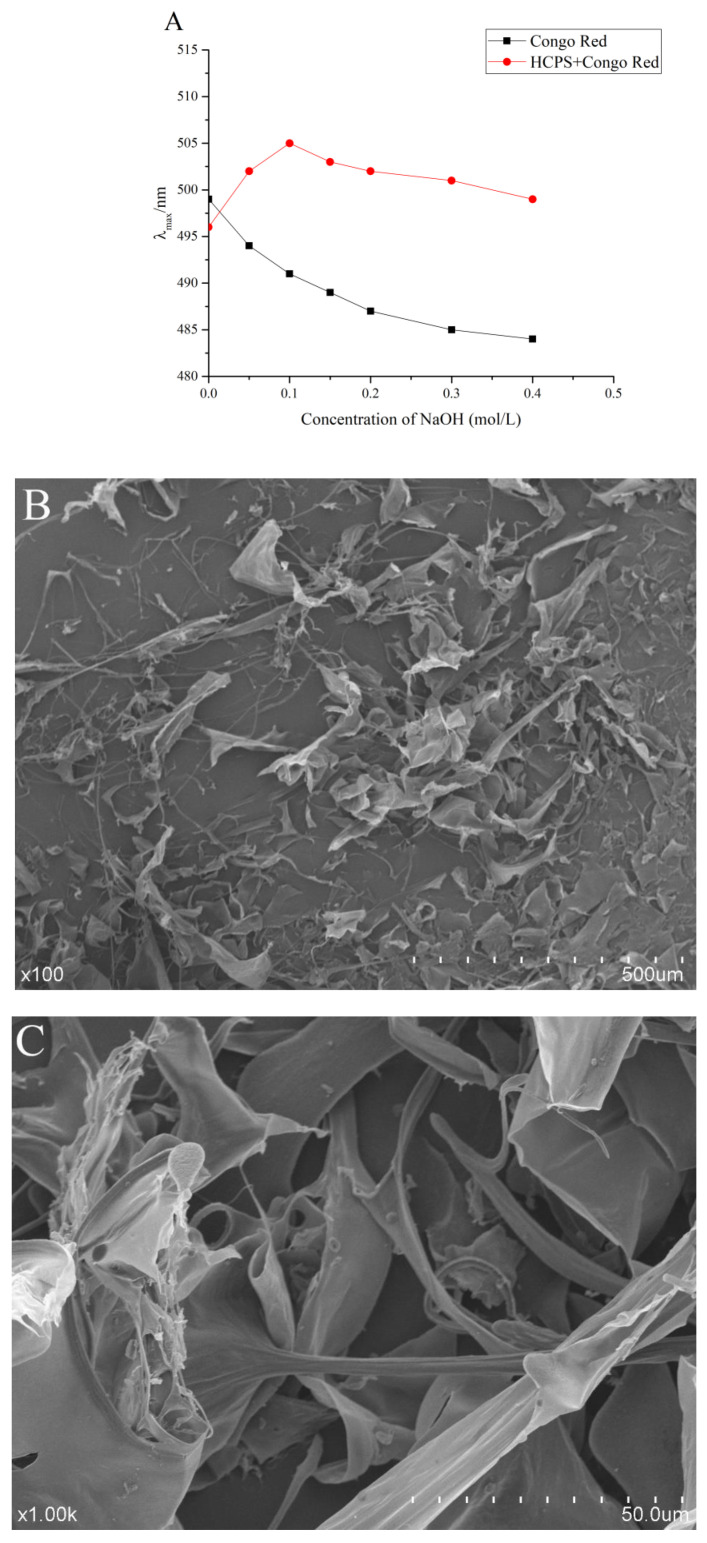
(**A**) Maximum absorption wavelength of Congo red and Congo red + HCPS at various concentrations of sodium hydroxide solution, (**B**) SEM images of HCPS (×100), (**C**) SEM images of HCPS (×1.00k).

**Figure 6 foods-11-03622-f006:**
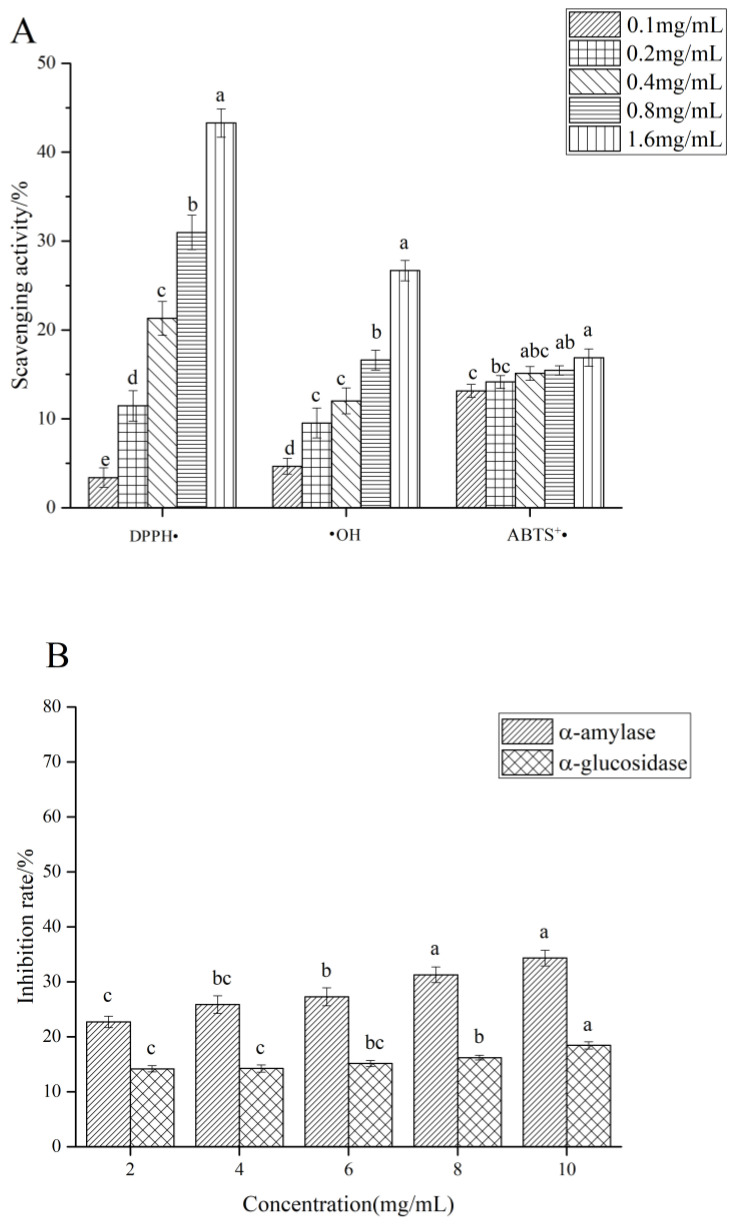
(**A**) The antioxidant activity in vitro of HCPS, (**B**) α-amylase inhibitory activity and α-glucosidase inhibitory activity of HCPS, (**C**) the antioxidant activity in vitro of Vc, (**D**) α-amylase inhibitory activity and α-glucosidase inhibitory activity of acarbose. Different lower-case letters were different significantly (*p* < 0.05).

**Table 1 foods-11-03622-t001:** Structural characterization and biological activities of *Houttuynia cordata* polysaccharides.

Polysaccharide Name	Molecular Weight	Monosaccharide Composition	Glycosidic Bond	Biological Activity	Reference
HCP	-	Man:Rha:GlcA:GalA:Glc:Xyl:Gal:Ara = 1.8:17.2:6.8:29.4:5.3:2.1:24:13.5	-	anti-oxidative	[19]
HP	43kDa	GalA:Gal:Glc:Xyl = 1.56:1.49:1.26:1.11	α-1,4-linked Gal*p*A, β-1,4-linked Gal*p*, β-1,4-linked Glc*p*, β-1,4-linked Xyl*p*	anti-viral	[23]
HCA4S1	21.7 kDa	Rha:GalA:Gal:Ara = 15.6:17.5:41.5:25.7	1,4-linked-α-d-GalA; 1,2,4-linked-α-L-Rha	anti-cancer	[24]
HC-PS1	274.53 kDa	Man, Rha, GlcA, GalA, Glc, Xyl, Gal, Ara	terminal Rha*p*; 1,5-linked Ara*f*; 1,3,6-linked and 1,4,6-linked Man*p*; 1,4-linked, 1,3-linked, 1,3,6-linked, 1,4,6-linked and 1,3,4,6-linked Glc*p*; 1,4-linked and 1,6-linked Gal*p*	immune regulation	[26]
HC-PS3	216.38 kDa		HC-PS3 was similar to HC-PS1. HC-PS3 did not have 1,3,6-linked Glc*p* but had additional 1,3,4-linked Man*p*		
HCP-2	60 kDa	GalA	1,4-linked-α-d-galacturonic acid	immune regulation	[22]
HCP	387 kDa	GalA, Gal, Glc	-	anti-inflammatory	[27]
HBHP-3	397.4 kDa	Rha:Ara:Glc:Gal:GalA = 16.0:12.6:4.6:18.1:15.6	→2)-α-l-Rha*p*-(1→, →4)-α-d-Gal*p*A-(1→ and →4)-β-D-Gal*p*-(1→	anti-inflammatory	[28]
HCP	-	Man, Rha, Glc, Xyl, Gal, Ara	-	Intestinal protection	[29]

**Table 2 foods-11-03622-t002:** Independent variables and levels in BBD.

Level	X_1_ Extraction Time (h)	X_2_ Extraction Temperature (°C)	X_3_ Solid-Liquid Ratio (g/mL)
−1	4	80	1:20
0	5	90	1:30
1	6	100	1:40

**Table 3 foods-11-03622-t003:** Response surface optimization design and results.

No.	X_1_	X_2_	X_3_	Extraction Yield (%)
1	1	0	1	2.11 ± 0.09
2	−1	0	1	1.70 ± 0.04
3	1	0	−1	2.08 ± 0.06
4	0	0	0	2.44 ± 0.08
5	0	0	0	2.42 ± 0.04
6	0	0	0	2.12 ± 0.04
7	0	−1	1	1.64 ± 0.02
8	1	−1	0	1.59 ± 0.08
9	0	1	−1	2.01 ± 0.07
10	−1	0	−1	1.66 ± 0.09
11	1	1	0	2.12 ± 0.06
12	−1	−1	0	1.67 ± 0.03
13	0	−1	−1	1.82 ± 0.01
14	0	0	0	2.28 ± 0.09
15	0	0	0	2.51 ± 0.04
16	0	1	1	2.05 ± 0.07
17	−1	1	0	1.85 ± 0.08

**Table 4 foods-11-03622-t004:** Analysis of variance of response surface results.

Source	Sum of Squares	DF	Mean Square	*F*-Value	*p*-Value
Model	1.27	9	0.14	6.39	0.0115 *
X_1_	0.13	1	0.13	5.89	0.0456 *
X_2_	0.21	1	0.21	9.72	0.0169 *
X_3_	0.001	1	0.001	0.028	0.8724
X_1_X_2_	0.031	1	0.031	1.39	0.2774
X_1_X_3_	0.00002	1	0.00002	0.001	0.9741
X_2_X_3_	0.0012	1	0.0012	0.55	0.4832
X_1_^2^	0.31	1	0.31	13.85	0.0074 **
X_2_^2^	0.32	1	0.32	14.63	0.0065 **
X_3_^2^	0.16	1	0.16	7.40	0.0297 *
Residual	0.15	7	0.022		
Lack of Fit	0.058	3	0.019	0.81	0.5526
Pure Error	0.096	4	0.024		
Cor Total	1.43	16			

Significantly different, * *p* < 0.05, ** *p* < 0.01.

**Table 5 foods-11-03622-t005:** Yield model of HCPS.

Model	R	R^2^	Adjusted R^2^	Standard Estimation Error
1	0.683 ^a^	0.466	0.425	0.16057
2	0.839 ^b^	0.703	0.654	0.12461

^a^ Predictive variable: (constant), temperature. ^b^ Predictive variables: (constant), temperature, time.

**Table 6 foods-11-03622-t006:** ANOVA analysis of yield of HCPS.

Model		Sum of Squares	DF	Mean Square	*F*	Sig.
1	Regression	0.292	1	0.292	11.342	0.005 ^b^
	Residual	0.335	13	0.026		
	Cor Total	0.628	14			
2	Regression	0.441	2	0.221	14.209	0.001 ^c^
	Residual	0.186	12	0.016		
	Cor Total	0.628	14			

^b^ Predictive variable: (constant), temperature. ^c^ Predictive variables: (constant), temperature, time.

**Table 7 foods-11-03622-t007:** The yield model coefficients of HCPS.

Model		Coefficient of Non-Standardization		Standardization Coefficient		Collinearity Statistics		
		B	Standard Error	Beta	t	Sig.	Tolerance	VIF
1	(Constant)	0.088	0.304		0.290	0.776		
	Temperature	0.018	0.004	0.794	4.705	0.000	1.000	1.000
2	(Constant)	−0.375	0.354		−1.059	0.311		
	Temperature	0.017	0.004	0.683	4.340	0.001	1.000	1.000
	Time	0.122	0.039	0.487	3.096	0.009	1.000	1.000

**Table 8 foods-11-03622-t008:** Pearson correlation analysis between factors and extraction yield of HCPS.

	Extraction Temperature	Extraction Time	Solid-Liquid Ratio
R	0.913 *	0.875	0.260
Sig. (two-sided)	0.031	0.052	0.673

Significantly different, * *p* < 0.05 (double tail).

## Data Availability

Data is contained within the article.

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
