# Peer review of "Structural Characterization and Biological Activity of Polysaccharides from Stems of Houttuynia cordata"

_foods, 2022, doi:10.3390/foods11223622_

Round 1
Reviewer 1 Report
1.The experimental design reported in Table1, lack of consistence: only few conditions have been investigated (3 conditions for 3 parameters) for a deeper RSM experiment further conditions are commonly required and may be un-necessary in this form. All RSM analysis performed I think it's redundant
2. Line 344: seems that the observed band may be attributable more to proteins content (280 nm circa) than nucleic acid (260 nm).
3. Structure characterization: Several and correct analysis have been performed, but they lack of details. Please provide further comments on the polysaccharide structure
4. FT-IR. Did you observe any band any band at 840cm-1 circa ascribable to C-O-S stretching? if is a sulphatate polysaccharide must be present and the quantitication of sulfate groups may be investigated
5. The C-NMR spectra is unnecessary in this form, please provide further details and better analysis or remove it
6. Please provide a better conclusion details
−1
Reviewer 2 Report
The manuscript provides some interesting data on the isolation and characterization analysis of polysaccharides from stems of Houttuynia cordata. The authors failed to address the novelty and importance of this manuscript. I have a few comments:
1. Introduction: please provides more details for 'The activity of H. cordata polysaccharide has been extensively studies'
2. Introduction: the authors mentioned 'Nevertheless, 70 the effect of H. cordata polysaccharides on α-glucosidase activity is unknown. ', and wrote 'This study determined the optimal conditions for extracting polysaccharides from 476 Houttuynia cordata stems (HCPS) using response surface methodology and Sephadex G- 477 200.' in Conclusion. It is hard to understand the real purpose of this research. Please reorganize the relevant parts.
3. Methods: References for sections 2.9 and 2.12?
4. Results: Where are the discussions of results? The manuscript is too technical, without sufficient discussion of their own data and the comparison with previous studies.
Round 2
Reviewer 1 Report
-Accept after minor revision
Reviewer 2 Report
I have no more comments now.